# Pointwise Binary Classification with Pairwise Confidence Comparisons

## Abstract

Ordinary (pointwise) binary classification aims to learn a binary classifier from pointwise labeled data. However, such pointwise labels may not be directly accessible due to privacy, confidentiality, or security considerations. In this case, can we still learn an accurate binary classifier? This paper proposes a novel setting, namely *pairwise comparison (Pcomp) classification*, where we are given only pairs of unlabeled data that we know one is more likely to be positive than the other, instead of pointwise labeled data. Compared with pointwise labels, pairwise comparisons are easier to collect, and Pcomp classification is useful for subjective classification tasks. To solve this problem, we present a mathematical formulation for the generation process of pairwise comparison data, based on which we exploit an *unbiased risk estimator* (URE) to train a binary classifier by *empirical risk minimization* and establish an *estimation error bound*. We first prove that a URE can be derived and improve it using *correction functions*. Then, we start from the *noisy-label learning* perspective to introduce a *progressive* URE and improve it by imposing *consistency regularization*. Finally, experiments validate the effectiveness of our proposed solutions for Pcomp classification.

## 1 Introduction

Traditional supervised learning techniques have achieved great advances, while they are demanding for precisely labeled data. In many real-world scenarios, it may be too difficult to collect such data. To alleviate this issue, a large number of weakly supervised learning problems (Zhou, 2018) have been extensively studied, including *semi-supervised learning* (Zhu & Goldberg, 2009; Niu et al., 2013; Sakai et al., 2018), *multi-instance learning* (Zhou et al., 2009; Sun et al., 2016; Zhang & Zhou, 2017), *noisy-label learning* (Han et al., 2018; Xia et al., 2019; Wei et al., 2020), *partial-label learning* (Zhang et al., 2017; Feng et al., 2020b; Lv et al., 2020), *complementary-label learning* (Ishida et al., 2017; Yu et al., 2018; Ishida et al., 2019; Feng et al., 2020a), *positive-unlabeled classification* (Gong et al., 2019), *positive-confidence classification* (Ishida et al., 2018), *similar-unlabeled classification* (Bao et al., 2018), *unlabeled-unlabeled classification* (Lu et al., 2019; 2020), and *triplet classification* (Cui et al., 2020).

This paper considers another novel weakly supervised learning setting called *pairwise comparison (Pcomp) classification*, where we aim to perform pointwise binary classification with only *pairwise comparison data*, instead of pointwise labeled data. A pairwise comparison $(\boldsymbol{x}, \boldsymbol{x}')$ represents that the instance $\boldsymbol{x}$ has a larger confidence of belonging to the positive class than the instance $\boldsymbol{x}'$. Such weak supervision (pairwise confidence comparison) could be much easier for people to collect than full supervision (pointwise label) in practice, especially for applications on sensitive or private matters. For example, it may be difficult to collect sensitive or private data with pointwise labels, as asking for the true labels could be prohibited or illegal. In this case, it could be easier for people to collect other weak supervision like the comparison information between two examples.

It is also advantageous to consider pairwise confidence comparisons in pointwise binary classification with class overlapping, where the labeling task becomes difficult, and even experienced labelers may provide wrong pointwise labels. Let us denote the labeling standard of a labeler as $\tilde{p}(y|\boldsymbol{x})$ and assume that an instance $\boldsymbol{x}_1$ is more positive than another instance $\boldsymbol{x}_2$. Facing the difficult labeling task, different labelers may hold different labeling standards, $\tilde{p}(y = +1|\boldsymbol{x}_1) > \tilde{p}(y = +1|\boldsymbol{x}_2) > 1/2, \tilde{p}(y = +1|\boldsymbol{x}_1) > 1/2 > \tilde{p}(y = +1|\boldsymbol{x}_2)$, and $1/2 > \tilde{p}(y = +1|\boldsymbol{x}_1) > \tilde{p}(y = +1|\boldsymbol{x}_2)$, thereby

providing different pointwise labels: $(+1, +1)$, $(+1, -1)$, $(-1, -1)$. We can find that different labelers may provide inconsistent pointwise labels, while pairwise confidence comparisons are unanimous and accurate. One may argue that we could aggregate multiple labels of the same instance using crowdsourcing learning methods (Whitehill et al., 2009; Raykar et al., 2010). However, as not every instance will be labeled by multiple labelers, it is not always applicable to crowdsourcing learning methods. Therefore, our proposed Pcomp classification is useful in this case.

Our contributions in this paper can be summarized as follows:

- We propose Pcomp classification, a novel weakly supervised learning setting, and present a mathematical formulation for the generation process of pairwise comparison data.

- We prove that an *unbiased risk estimator* (URE) can be derived, propose an *empirical risk minimization* (ERM) based method, and present an improvement using correction functions (Lu et al., 2020) for alleviating overftting when complex models are used.

- We start from the noisy-label learning perspective to introduce the *RankPruning* method (Northcutt et al., 2017) that holds a *progressive* URE for solving our proposed Pcomp classification problem and improve it by imposing *consistency regularization*.

- We experimentally demonstrate the effectiveness of our proposed solutions for Pcomp classification.

## 2 PRELIMINARIES

Binary classification with pairwise comparisons and extra pointwise labels has been studied (Xu et al., 2017; Kane et al., 2017). Our paper focuses on a more challenging problem where only pairwise comparison examples are provided. Unlike previous studies (Xu et al., 2017; Kane et al., 2017) that leverage some pointwise labels to differentiate the labels of pairwise comparisons, our methods are purely based on ERM with only pairwise comparisons. In the next, we briefly introduce some notations and review the related problem formulations of binary classification, positive-unlabeled classification, and unlabeled-unlabeled classification.

**Binary Classification.**   Since our paper focuses on how to train a binary classifier from pairwise comparison data, we first review the problem formulation of binary classification. Let the feature space be $\mathcal{X}$ and the label space be $\mathcal{Y} = \{+1, -1\}$. Suppose the collected dataset is denoted by $\mathcal{D} = \{(\boldsymbol{x}_i, y_i)\}_{i=1}^n$ where each example $(\boldsymbol{x}_i, y_i)$ is independently sampled from the joint distribution with density $p(\boldsymbol{x}, y)$, which includes an instance $\boldsymbol{x}_i \in \mathcal{X}$ and a label $y_i \in \mathcal{Y}$. The goal of binary classification is to train an optimal classifier $f : \mathcal{X} \mapsto \mathbb{R}$ by minimizing the following expected classification risk:

$$R(f) = \mathbb{E}_{p(\boldsymbol{x},y)}\big[\ell(f(\boldsymbol{x}), y)\big] = \pi_+ \mathbb{E}_{p_+(\boldsymbol{x})}\big[\ell(f(\boldsymbol{x}), +1)\big] + \pi_- \mathbb{E}_{p_-(\boldsymbol{x})}\big[\ell(f(\boldsymbol{x}), -1)\big], \quad (1)$$

where $\ell : \mathbb{R} \times \mathcal{Y} \mapsto \mathbb{R}_+$ denotes a binary loss function, $\pi_+ := p(y = +1)$ (or $\pi_- := p(y = -1)$) denotes the *positive* (or *negative*) *class prior probability*, and $p_+(\boldsymbol{x}) := p(\boldsymbol{x}|y = +1)$ (or $p_-(\boldsymbol{x}) := p(\boldsymbol{x}|y = -1)$) denotes the *class-conditional probability density* of the positive (or negative) data. ERM approximates the expectations over $p_+(x)$ and $p_-(x)$ by the empirical averages of positive and negative data and the empirical risk is minimized with respect to the classifier $f$.

**Positive-Unlabeled (PU) Classification.**   In some real-world scenarios, it may be difficult to collect negative data, and only positive (P) and unlabeled (U) data are available. PU classification aims to train an effective binary classifier in this weakly supervised setting. Previous studies (du Plessis et al., 2014; 2015; Kiryo et al., 2017) showed that the classification risk $R(f)$ in Eq. (1) can be rewritten only in terms of positive and unlabeled data as

$$R(f) = R_{\mathrm{PU}}(f) = \pi_+ \mathbb{E}_{p_+(\boldsymbol{x})}\big[\ell(f(\boldsymbol{x}), +1) - \ell(f(\boldsymbol{x}), -1)\big] + \mathbb{E}_{p(\boldsymbol{x})}\big[\ell(f(\boldsymbol{x}), -1)\big], \quad (2)$$

where $p(\boldsymbol{x}) = \pi_+ p_+(\boldsymbol{x}) + \pi_- p_-(\boldsymbol{x})$ denotes the probability density of unlabeled data. This risk expression immediately allows us to employ ERM in terms of positive and unlabeled data.

**Unlabeled-Unlabeled (UU) Classification.**   The recent studies (Lu et al., 2019; 2020) showed that it is possible to train a binary classifier only from two unlabeled datasets with different class priors.

Lu et al. (2019) showed that the classification risk can be rewritten as

$$R(f) = R_{\mathrm{UU}}(f) = \mathbb{E}_{p_{\mathrm{tr}}(\boldsymbol{x})}\left[\frac{(1-\theta')\pi_+}{\theta - \theta'}\ell(f(\boldsymbol{x}), +1) - \frac{\theta'(1-\pi_+)}{\theta - \theta'}\ell(f(\boldsymbol{x}), -1)\right]$$
$$+ \mathbb{E}_{p_{\mathrm{tr}'}(\boldsymbol{x}')}\left[\frac{\theta(1-\pi_+)}{\theta - \theta'}\ell(f(\boldsymbol{x}'), -1) - \frac{(1-\theta)\pi_+}{\theta - \theta'}\ell(f(\boldsymbol{x}'), +1)\right], \quad (3)$$

where $\theta$ and $\theta'$ are different class priors of two unlabeled datasets, and $p_{\mathrm{tr}}(\boldsymbol{x})$ and $p_{\mathrm{tr}'}(\boldsymbol{x}')$ are the densities of two datasets of unlabeled data, respectively. This risk expression immediately allows us to employ ERM only from two sets of unlabeled data. For $R_{\mathrm{UU}}(f)$ in Eq. (3), if we set $\theta = 1$, $\theta' = \pi_+$, and replace $p_{\mathrm{tr}}(\boldsymbol{x})$ and $p_{\mathrm{tr}'}(\boldsymbol{x}')$ by $p_+(\boldsymbol{x})$ and $p(\boldsymbol{x})$ respectively, then we can recover $R_{\mathrm{PU}}(f)$ in Eq. (2). Therefore, UU classification could be taken as a generalized framework of PU classification in terms of URE. Besides, Eq. (3) also recovers a complicated URE of similar-unlabeled classification (Bao et al., 2018) by setting $\theta = \pi_+$ and $\theta' = \pi_+^2/(2\pi_+^2 - 2\pi_+ + 1)$.

To solve our proposed Pcomp classification problem, we will present a mathematical formulation for the generation process of pairwise comparison data, based on which we will explore two UREs to train a binary classifier by ERM and establish the corresponding *estimation error bounds*.

## 3 DATA GENERATION PROCESS

In order to derive UREs for performing ERM, we first formulate the underlying generation process of pairwise comparison data[1], which consists of pairs of unlabeled data that we know which one is more likely to be positive. Suppose the provided dataset is denoted by $\widetilde{\mathcal{D}} = \{(\boldsymbol{x}_i, \boldsymbol{x}_i')\}_{i=1}^n$ where $(\boldsymbol{x}_i, \boldsymbol{x}_i')$ (with their unknown true labels $(y_i, y_i')$) is expected to satisfy $p(y_i = +1|\boldsymbol{x}_i) > p(y_i' = +1|\boldsymbol{x}_i')$.

It is clear that we could easily collect pairwise comparison data if the positive confidence (i.e., $p(y = +1|\boldsymbol{x})$) of each instance could be obtained. However, such information is much harder to obtain than class labels in real-world scenarios. Therefore, unlike some studies (Ishida et al., 2018; Shinoda et al., 2020) that assume the positive confidence of each instance is provided by the labeler, we only assume that the labeler has access to the labels of training data. Specifically, we adopt the assumption (Cui et al., 2020) that weakly supervised examples are first sampled from the true data distribution, but the labels are only accessible to the labeler. Then, the labeler would provide us weakly supervised information (i.e., pairwise comparison information) according to the labels of sampled data pairs. That is, for any pair of unlabeled data $(\boldsymbol{x}, \boldsymbol{x}')$, the labeler would tell us whether $(\boldsymbol{x}, \boldsymbol{x}')$ could be collected as a pairwise comparison for Pcomp classification, based on the labels $(y, y')$ rather than the positive confidences $(p(y = +1|\boldsymbol{x}), p(y = +1|\boldsymbol{x}'))$.

Now, the question becomes: how does the labeler consider $(\boldsymbol{x}, \boldsymbol{x}')$ as a pairwise comparison for Pcomp classification, in terms of the labels $(y, y')$? As shown in our previous example of binary classification with class overlapping, we could infer that the labels $(y, y')$ of our required pairwise comparison data $(\boldsymbol{x}, \boldsymbol{x}')$ for Pcomp classification can only be one of the three cases $\{(+1, -1), (+1, +1), (-1, -1)\}$, because the condition $p(y = +1|\boldsymbol{x}) \geq p(y' = +1|\boldsymbol{x}')$ is definitely violated if $(y, y') = (-1, +1)$. Therefore, we assume that the labeler would take $(\boldsymbol{x}, \boldsymbol{x}')$ as a pairwise comparison example in the dataset $\widetilde{\mathcal{D}}$, if the labels $(y, y')$ of $(\boldsymbol{x}, \boldsymbol{x}')$ belong to the above three cases. It is also worth noting that for a pair of data $(\boldsymbol{x}, \boldsymbol{x}')$ with labels $(y, y') = (-1, +1)$, the labeler would take $(\boldsymbol{x}', \boldsymbol{x})$ as a pairwise comparison example. Because by exchanging the positions of $(\boldsymbol{x}, \boldsymbol{x}')$, $(\boldsymbol{x}', \boldsymbol{x})$ would be associated with labels $(+1, -1)$, which belong to the three cases. In summary, we assume that pairwise comparison data are sampled from those pairs of data whose labels belong to the three cases $\{(+1, -1), (+1, +1), (-1, -1)\}$. Based on the above described generation process of pairwise comparison data, we have the following theorem.

**Theorem 1.** *According to the generation process of pairwise comparison data described above, let*

$$\widetilde{p}(\boldsymbol{x}, \boldsymbol{x}') = \frac{q(\boldsymbol{x}, \boldsymbol{x}')}{\pi_+^2 + \pi_-^2 + \pi_+\pi_-}, \quad (4)$$

*where $q(\boldsymbol{x}, \boldsymbol{x}') = \pi_+^2 p_+(\boldsymbol{x})p_+(\boldsymbol{x}') + \pi_-^2 p_-(\boldsymbol{x})p_-(\boldsymbol{x}') + \pi_+\pi_- p_+(\boldsymbol{x})p_-(\boldsymbol{x}')$. Then we have $\widetilde{\mathcal{D}} = \{(\boldsymbol{x}_i, \boldsymbol{x}_i')\}_{i=1}^n \overset{\text{i.i.d.}}{\sim} \widetilde{p}(\boldsymbol{x}, \boldsymbol{x}')$.*

---

[1]In contrast to Xu et al. (2019) and Xu et al. (2020) which utilized pairwise comparison data to solve the regression problem, we focus on binary classification.

The proof is provided in Appendix A. Theorem 1 provides an explicit expression of the probability density of pairwise comparison data.

Next, we would like to extract pointwise information from pairwise information, since our goal is to perform pointwise binary classification. Let $\widetilde{\pi} = \pi_+^2 + \pi_-^2 + \pi_+\pi_- = \pi_+ + \pi_-^2 = \pi_+^2 + \pi_-$ and we denote the pointwise data collected from $\widetilde{\mathcal{D}} = \{(\boldsymbol{x}_i, \boldsymbol{x}_i')\}_{i=1}^n$ by breaking the pairwise comparison relation as $\widetilde{\mathcal{D}}_+ = \{\boldsymbol{x}_i\}_{i=1}^n$ and $\widetilde{\mathcal{D}}_- = \{\boldsymbol{x}_i'\}_{i=1}^n$. Then we can obtain the following theorem.

**Theorem 2.** *Pointwise examples in $\widetilde{\mathcal{D}}_+$ and $\widetilde{\mathcal{D}}_-$ are independently drawn from $\widetilde{p}_+(\boldsymbol{x})$ and $\widetilde{p}_-(\boldsymbol{x}')$, where*

$$\widetilde{p}_+(\boldsymbol{x}) = \frac{\pi_+}{\pi_-^2 + \pi_+}p_+(\boldsymbol{x}) + \frac{\pi_-^2}{\pi_-^2 + \pi_+}p_-(\boldsymbol{x}), \quad \widetilde{p}_-(\boldsymbol{x}') = \frac{\pi_+^2}{\pi_+^2 + \pi_-}p_+(\boldsymbol{x}') + \frac{\pi_-}{\pi_+^2 + \pi_-}p_-(\boldsymbol{x}').$$

The proof is provided in Appendix B. Theorem 2 shows the relationships between the pointwise densities and the class-conditional densities. Besides, it indicates that from pairwise comparison data, we can essentially obtain examples that are independently drawn from $\widetilde{p}_+(\boldsymbol{x})$ and $\widetilde{p}_-(\boldsymbol{x}')$.

## 4    THE PROPOSED METHODS

In this section, we explore two UREs to train a binary classifier by ERM from only pairwise comparison data with the above generation process.

### 4.1    CORRECTED PCOMP CLASSIFICATION

As shown in Eq. (1), the classification risk $R(f)$ could be separately expressed as the expectations over $p_+(\boldsymbol{x})$ and $p_-(\boldsymbol{x})$. Although we do not have access to the two class-conditional densities $p_+(\boldsymbol{x})$ and $p_-(\boldsymbol{x})$, we can represent them by our introduced pointwise densities $\widetilde{p}_+(\boldsymbol{x})$ and $\widetilde{p}_-(\boldsymbol{x})$.

**Lemma 1.** *We can express $p_+(\boldsymbol{x})$ and $p_-(\boldsymbol{x})$ in terms of $\widetilde{p}_+(\boldsymbol{x})$ and $\widetilde{p}_+(\boldsymbol{x})$ as*

$$p_+(\boldsymbol{x}) = \frac{1}{\pi_+}\big(\widetilde{p}_+(\boldsymbol{x}) - \pi_-\widetilde{p}_-(\boldsymbol{x})\big), \quad p_-(\boldsymbol{x}) = \frac{1}{\pi_-}\big(\widetilde{p}_-(\boldsymbol{x}) - \pi_+\widetilde{p}_+(\boldsymbol{x})\big).$$

The proof is provided in Appendix C. As a result of Lemma 1, we can express the classification risk $R(f)$ using only pairwise comparison data sampled from $\widetilde{p}_+(\boldsymbol{x})$ and $\widetilde{p}_-(\boldsymbol{x})$.

**Theorem 3.** *The classification risk $R(f)$ can be equivalently expressed as*

$$R_{\mathrm{PC}}(f) = \mathbb{E}_{\widetilde{p}_+(\boldsymbol{x})}\big[\ell(f(\boldsymbol{x}), +1) - \pi_+\ell(f(\boldsymbol{x}), -1)\big] + \mathbb{E}_{\widetilde{p}_-(\boldsymbol{x}')}\big[\ell(f(\boldsymbol{x}'), -1) - \pi_-\ell(f(\boldsymbol{x}'), +1)\big]. \tag{5}$$

The proof is provided in Appendix D. In this way, we could train a binary classifier by minimizing the following empirical approximation of $R_{\mathrm{PC}}(f)$:

$$\widehat{R}_{\mathrm{PC}}(f) = \frac{1}{n}\sum_{i=1}^n \Big(\ell(f(\boldsymbol{x}_i), +1) - \pi_+\ell(f(\boldsymbol{x}_i), -1) + \ell(f(\boldsymbol{x}_i'), -1) - \pi_-\ell(f(\boldsymbol{x}_i'), +1)\Big). \tag{6}$$

**Estimation Error Bound.**    Here, we establish an estimation error bound for the proposed URE. Let $\mathcal{F} = \{f : \mathcal{X} \mapsto \mathbb{R}\}$ be the model class, $\widehat{f}_{\mathrm{PC}} = \arg\min_{f \in \mathcal{F}} \widehat{R}_{\mathrm{PC}}(f)$ be the empirical risk minimizer, and $f^\star = \arg\min_{f \in \mathcal{F}} R(f)$ be the true risk minimizer. Let $\widetilde{\mathfrak{R}}_n^+(\mathcal{F})$ and $\widetilde{\mathfrak{R}}_n^-(\mathcal{F})$ be the *Rademacher complexities* (Bartlett & Mendelson, 2002) of $\mathcal{F}$ with sample size $n$ over $\widetilde{p}_+(\boldsymbol{x})$ and $\widetilde{p}_-(\boldsymbol{x})$ respectively.

**Theorem 4.** *Suppose the loss function $\ell$ is $\rho$-Lipschitz with respect to the first argument ($0 \leq \rho \leq \infty$), and all functions in the model class $\mathcal{F}$ are bounded, i.e., there exists a positive constant $C_{\mathrm{b}}$ such that $\|f\| \leq C_{\mathrm{b}}$ for any $f \in \mathcal{F}$. Let $C_\ell := \sup_{z \leq C_{\mathrm{b}}, t=\pm 1} \ell(z, t)$. Then for any $\delta > 0$, with probability at least $1 - \delta$, we have*

$$R(\widehat{f}_{\mathrm{PC}}) - R(f^\star) \leq (1 + \pi_+)4\rho\widetilde{\mathfrak{R}}_n^+(\mathcal{F}) + (1 + \pi_-)4\rho\widetilde{\mathfrak{R}}_n^-(\mathcal{F}) + 6C_\ell\sqrt{\frac{\log\frac{8}{\delta}}{2n}}.$$

The proof is provided in Appendix E. Theorem 4 shows that our proposed method is consistent, i.e., as $n \to \infty$, $R(\widehat{f}_{\mathrm{PC}}) \to R(f^\star)$, since $\widetilde{\mathfrak{R}}_n^+(\mathcal{F}), \widetilde{\mathfrak{R}}_n^-(\mathcal{F}) \to 0$ for all parametric models with a bounded norm such as deep neural networks trained with weight decay (Golowich et al., 2017; Lu et al., 2019). Besides, $\widetilde{\mathfrak{R}}_n^+(\mathcal{F})$ and $\widetilde{\mathfrak{R}}_n^-(\mathcal{F})$ can be normally bounded by $C_\mathcal{F}/\sqrt{n}$ for a positive constant $C_\mathcal{F}$. Hence, we can further see that the convergence rate is $\mathcal{O}_p(1/\sqrt{n})$ where $\mathcal{O}_p$ denotes the order in probability. This order is the optimal parametric rate for ERM without additional assumptions (Mendelson, 2008).

**Relation to UU Classification.** It is worth noting that the URE of UU classification $R_{\mathrm{UU}}(f)$ is quite general for binary classification with weak supervision. Hence we also would like to show the relationships between our proposed estimator $R_{\mathrm{PC}}(f)$ and $R_{\mathrm{UU}}(f)$. We demonstrate by the following corollary that under some conditions, $R_{\mathrm{UU}}(f)$ is equivalent to $R_{\mathrm{PC}}(f)$.

**Corollary 1.** *By setting $p_{\mathrm{tr}} = \widetilde{p}_+(\boldsymbol{x})$, $p'_{\mathrm{tr}} = \widetilde{p}_-(\boldsymbol{x})$, $\theta = \pi_+/(1 - \pi_+ + \pi_+^2)$, and $\theta' = \pi_+^2/(1 - \pi_+ + \pi_+^2)$, Eq. (3) is equivalent to Eq. (5), which means that $R_{\mathrm{UU}}(f)$ is equivalent to $R_{\mathrm{PC}}(f)$.*

We omit the proof of Corollary 1 since it is straightforward to derive Eq. (5) from Eq. (3) by inserting the required notations.

**Empirical Risk Correction.** As shown in Lu et al. (2020), directly minimizing $\widehat{R}_{\mathrm{PC}}(f)$ would suffer from overfitting when complex models are used due to the negative risk issue. More specifically, since negative terms are included in Eq. (6), the empirical risk can be negative even though the original true risk can never be negative. To ease this problem, they wrapped the terms in $\widehat{R}_{\mathrm{UU}}(f)$ that cause a negative empirical risk by certain *consistent correction functions* such as the rectified linear unit (ReLU) function $g(z) = \max(0, z)$ and absolute value function $g(z) = |z|$. This solution could also be applied to $\widehat{R}_{\mathrm{PC}}$. In this way, we could obtain the following corrected empirical risk estimator:

$$\widehat{R}_{\mathrm{cPC}}(f) = g\Big(\frac{1}{n}\sum\nolimits_{i=1}^n \big(\ell(f(\boldsymbol{x}_i), +1) - \pi_-\ell(f(\boldsymbol{x}'_i), +1)\big)\Big)$$
$$+ g\Big(\frac{1}{n}\sum\nolimits_{i=1}^n \big(\ell(f(\boldsymbol{x}'_i), -1) - \pi_+\ell(f(\boldsymbol{x}_i), -1)\big)\Big). \quad (7)$$

## 4.2 PROGRESSIVE PCOMP CLASSIFICATION

Here, we start from the noisy-label learning perspective to solve the Pcomp classification problem. Intuitively, we could simply perform binary classification by regarding the data from $\widetilde{p}_+(\boldsymbol{x})$ as (noisy) positive data and the data from $\widetilde{p}_-(\boldsymbol{x})$ as (noisy) negative data. However, this naive solution could be inevitably affected by noisy labels. In this scenario, we denote the noise rates as $\rho_- = p(\widetilde{y} = +1|y = -1)$ and $\rho_+ = p(\widetilde{y} = -1|y = +1)$, where $\widetilde{y}$ is the observed (noisy) label and $y$ is the true label, and the inverse noise rates as $\phi_+ = p(y = -1|\widetilde{y} = +1)$ and $\phi_- = p(y = +1|\widetilde{y} = -1)$. According to the defined generation process of pairwise comparison data, we have the following theorem.

**Theorem 5.** *The following equalities hold:*

$$\phi_+ = \frac{\pi_-^2}{\pi_+^2 + \pi_-^2 + \pi_+\pi_-}, \quad \phi_- = \frac{\pi_+^2}{\pi_+^2 + \pi_-^2 + \pi_+\pi_-},$$
$$\rho_+ = \frac{\pi_+}{2(\pi_+^2 + \pi_-^2 + \pi_+\pi_-)}, \quad \rho_- = \frac{\pi_-}{2(\pi_+^2 + \pi_-^2 + \pi_+\pi_-)}.$$

The proof is provided in Appendix F.

Theorem 5 shows that the noise rates can be obtained if we regard the Pcomp classification problem as the noisy-label learning problem. With known noise rates for noisy-label learning, it was shown (Natarajan et al., 2013; Northcutt et al., 2017) that a URE could be derived. Here, we adopt the RankPruning method (Northcutt et al., 2017) because it holds a progressive URE by selecting confident examples using the learning model and achieves state-of-the-art performance. Specifically, we denote by the dataset composed of all the observed positive data $\widetilde{\mathcal{P}}$, i.e., $\widetilde{\mathcal{P}} = \{\boldsymbol{x}_i\}_{i=1}^n$, where $\boldsymbol{x}_i$ is independently sampled from $\widetilde{p}_+(\boldsymbol{x})$. Similarly, the dataset composed of all the observed negative data is denoted by $\widetilde{\mathcal{N}}$, i.e., $\widetilde{\mathcal{N}} = \{\boldsymbol{x}'_i\}_{i=1}^n$, where $\boldsymbol{x}'_i$ is independently sampled from $\widetilde{p}_-(\boldsymbol{x}')$. Then,

confident examples will be selected from $\widetilde{\mathcal{P}}$ and $\widetilde{\mathcal{N}}$ by ranking the outputs of the model $f$. We denote the selected positive data from $\widetilde{\mathcal{P}}$ as $\widetilde{\mathcal{P}}_{\text{sel}}$, and the selected negative data from $\widetilde{\mathcal{N}}$ as $\widetilde{\mathcal{N}}_{\text{sel}}$:

$$\widetilde{\mathcal{P}}_{\text{sel}} = \underset{\mathcal{P}:|\mathcal{P}|=(1-\phi_+)|\widetilde{\mathcal{P}}|}{\arg\max} \sum_{\boldsymbol{x}\in\{\mathcal{P}\cap\widetilde{\mathcal{P}}\}} f(\boldsymbol{x}), \quad \widetilde{\mathcal{N}}_{\text{sel}} = \underset{\mathcal{N}:|\mathcal{N}|=(1-\phi_-)|\widetilde{\mathcal{N}}|}{\arg\min} \sum_{\boldsymbol{x}\in\{\mathcal{N}\cap\widetilde{\mathcal{N}}\}} f(\boldsymbol{x}).$$

Then we show that if the model $f$ satisfies the *separability condition*, i.e., for any true positive instance $\boldsymbol{x}_{\text{p}}$ and for any true negative instance $\boldsymbol{x}_{\text{n}}$, we have $f(\boldsymbol{x}_{\text{p}}) > f(\boldsymbol{x}_{\text{n}})$. In other words, the model output of every true positive instance is always larger than that of every true negative instance, we could obtain a URE. We name it progressive URE, as the model $f$ is progressively optimized.

**Theorem 6** (Theorem 5 in Northcutt et al. (2017)). *Assume that the model $f$ satisfies the above separability condition, then the classification risk $R(f)$ can be equivalently expressed as*

$$R_{\text{pPC}}(f) = \mathbb{E}_{\widetilde{p}_+(\boldsymbol{x})}\Big[\frac{\ell(f(\boldsymbol{x}),+1)}{1-\rho_+}\mathbb{I}[\boldsymbol{x}\in\widetilde{\mathcal{P}}_{\text{sel}}]\Big] + \mathbb{E}_{\widetilde{p}_-(\boldsymbol{x}')}\Big[\frac{\ell(f(\boldsymbol{x}'),-1)}{1-\rho_-}\mathbb{I}[\boldsymbol{x}'\in\widetilde{\mathcal{N}}_{\text{sel}}]\Big],$$

*where $\mathbb{I}[\cdot]$ is the indicator function.*

In this way, we have the following empirical approximation of $R_{\text{pPC}}$:

$$\widehat{R}_{\text{pPC}}(f) = \frac{1}{n}\sum_{i=1}^{n}\Big(\frac{\ell(f(\boldsymbol{x}_i),+1)}{1-\rho_+}\mathbb{I}[\boldsymbol{x}_i\in\widetilde{\mathcal{P}}_{\text{sel}}] + \frac{\ell(f(\boldsymbol{x}'_i),-1)}{1-\rho_-}\mathbb{I}[\boldsymbol{x}'_i\in\widetilde{\mathcal{N}}_{\text{sel}}]\Big). \quad (8)$$

**Estimation Error Bound.** It worth noting that Northcutt et al. (2017) did not prove the learning consistency for the RankPruning method. Here, we establish an estimation error bound for this method, which guarantees the learning consistency. Let $\widehat{f}_{\text{pPC}} = \arg\min_{f\in\mathcal{F}} \widehat{R}_{\text{pPC}}(f)$ be the empirical risk minimizer of the RankPruning method, then we have the following theorem.

**Theorem 7.** *Suppose the loss function $\ell$ is $\rho$-Lipschitz with respect to the first argument ($0\leq\rho\leq\infty$), and all functions in the model class $\mathcal{F}$ are bounded, i.e., there exists a positive constant $C_{\text{b}}$ such that $\|f\|\leq C_{\text{b}}$ for any $f\in\mathcal{F}$. Let $C_\ell := \sup_{z\leq C_{\text{b}}, t=\pm 1}\ell(z,t)$. Then for any $\delta > 0$, with probability at least $1-\delta$, we have*

$$R(\widehat{f}_{\text{pPC}}) - R(f^\star) \leq \frac{2}{1-\rho_+}\Big(2\rho\widetilde{\mathfrak{R}}_n^+(\mathcal{F}) + C_\ell\sqrt{\frac{\log\frac{4}{\delta}}{2n}}\Big) + \frac{2}{1-\rho_-}\Big(2\rho\widetilde{\mathfrak{R}}_n^-(\mathcal{F}) + C_\ell\sqrt{\frac{\log\frac{4}{\delta}}{2n}}\Big).$$

The proof is provided in Appendix G. Theorem 7 shows that the above method is consistent and this estimation error bound also attains the optimal convergence rate without any additional assumption (Mendelson, 2008), as analyzed in Theorem 4.

**Regularization.** For the above RankPruning method, its URE is based on the assumption that the learning model could satisfy the separability condition. Thus, its performance heavily depends on the accuracy of the learning model. However, as the learning model is progressively updated, some of the selected confident examples may still contain label noise during the training process. As a result, the RankPruning method would be affected by incorrectly selected data. A straightforward improvement could be to improve the output quality of the learning model. Motivated by Mean Teacher used in semi-supervised learning (Tarvainen & Valpola, 2017), we also resort to a teacher model that is an exponential moving average of model snapshots, i.e., $\Theta'_t = \alpha\Theta'_{t-1} + (1-\alpha)\Theta_t$, where $\Theta'$ denotes the parameters of the teacher model, $\Theta$ denotes the parameters of the learning model, the subscript $t$ denotes the training step, and $\alpha$ is a smoothing coefficient hyper-parameter. Such a teacher model could guide the learning model to produce high-quality outputs. To learn from the teacher model, we leverage consistency regularization $\Omega(f) = \mathbb{E}_{\boldsymbol{x}}\big[\|f_{\Theta}(\boldsymbol{x}) - f_{\Theta'}(\boldsymbol{x})\|^2\big]$ (Laine & Aila, 2016; Tarvainen & Valpola, 2017) to make the learning model consistent with the teacher model for improving the RankPruning method.

## 5 EXPERIMENTS

In this section, we conduct experiments to evaluate the practical performance of our proposed methods on various datasets.

Table 1: Classification accuracy (mean±std) in percentage of each method on the four benchmark datasets with different class priors. The best performance is highlighted in bold.

| Class Prior | Methods | MNIST | Kuzushiji | Fashion | CIFAR-10 |
|---|---|---|---|---|---|
| $\pi_+ = 0.2$ | Noisy-Unbiased | 86.52±3.48 | 64.47±9.88 | 91.98±0.35 | 80.00±0.00 |
| | Binary-Biased | 27.80±2.38 | 58.54±1.13 | 43.27±9.25 | 49.87±4.38 |
| | RankPruning | 93.58±0.49 | 81.58±1.23 | 94.36±0.54 | 84.02±0.51 |
| | Pcomp-ABS | 89.83±1.49 | **84.66±0.56** | 91.29±1.69 | 82.56±0.75 |
| | Pcomp-ReLU | 93.39±0.71 | 83.76±0.99 | 94.07±0.49 | 81.16±0.67 |
| | Pcomp-Unbiased | 80.52±4.73 | 60.06±9.28 | 89.74±2.27 | 64.49±2.08 |
| | Pcomp-Teacher | **94.08±0.56** | 83.82±0.48 | **94.38±0.53** | **84.42±0.76** |
| $\pi_+ = 0.5$ | Noisy-Unbiased | 86.10±3.26 | 65.41±3.48 | 89.74±2.31 | 62.40±2.08 |
| | Binary-Biased | 54.10±2.42 | 60.75±0.54 | 45.76±1.81 | 48.36±3.13 |
| | RankPruning | 89.64±0.21 | 78.41±0.72 | **92.72±0.34** | **81.23±0.71** |
| | Pcomp-ABS | 85.90±0.30 | 74.29±1.42 | 92.18±0.90 | 70.71±0.90 |
| | Pcomp-ReLU | 87.81±1.08 | 73.88±0.72 | 92.13±1.33 | 74.51±2.26 |
| | Pcomp-Unbiased | 85.37±4.08 | 64.84±4.61 | 91.02±0.94 | 62.50±1.78 |
| | Pcomp-Teacher | **89.85±0.40** | **78.95±0.66** | 92.55±0.40 | 80.21±2.36 |
| $\pi_+ = 0.8$ | Noisy-Unbiased | 85.73±3.63 | 76.60±4.06 | 88.96±0.57 | 72.73±6.92 |
| | Binary-Biased | 27.12±2.80 | 55.72±1.50 | 46.74±2.19 | 38.59±9.98 |
| | RankPruning | 93.86±0.72 | 82.25±2.32 | 94.60±0.24 | 84.34±1.30 |
| | Pcomp-ABS | 88.06±1.60 | 82.96±0.54 | 91.69±1.67 | 82.87±0.59 |
| | Pcomp-ReLU | 93.63±1.03 | 83.17±1.38 | 93.31±1.34 | 81.40±0.59 |
| | Pcomp-Unbiased | 80.49±4.03 | 67.30±3.57 | 80.02±4.82 | 66.48±9.61 |
| | Pcomp-Teacher | **94.96±0.38** | **84.22±1.21** | **94.63±0.43** | **84.86±0.15** |

**Datasets.** We use four popular benchmark datasets, including MNIST (LeCun et al., 1998), Fashion-MNIST (Xiao et al., 2017), Kuzushiji-MNIST (Clanuwat et al., 2018), and CIFAR-10 (Krizhevsky et al., 2009). We train a multilayer perceptron (MLP) model with three hidden layers of width 300 and ReLU activation functions (Nair & Hinton, 2010) and batch normalization (Ioffe & Szegedy, 2015) on the first three datasets. We train ResNet-34 (He et al., 2016) on the CIFAR-10 dataset. We also use USPS and three datasets from the UCI machine learning repository (Blake & Merz, 1998) including Pendigits, Optdigits, and CNAE-9. We train a linear model on these datasets, since they are not large-scale datasets. The detailed descriptions of all used datasets with the corresponding models are provided in Appendix H. Since these datasets are specially used for multi-class classification, we manually transformed them into binary classification datasets (please see Appendix H for details). As we have shown in Theorem 2, the pairwise comparison examples can be equivalently transformed into pointwise examples, which are more convenient to generate. Therefore, we generate pointwise examples in experiments. Specifically, as Theorem 5 discloses the noise rates in our defined data generation process, we simply generate pointwise corrupted examples according to the noise rates.

**Methods.** For our proposed Pcomp classification problem, we propose the following methods: **Pcomp-Unbiased**, which denotes the proposed method that minimizes $\widehat{R}_{\mathrm{PC}}(f)$ in Eq. (6); **Pcomp-ReLU**, which denotes the proposed method that minimizes $\widehat{R}_{\mathrm{cPC}}(f)$ in Eq. (7) with the ReLU function; **Pcomp-ABS**, which denotes the proposed method that minimizes $\widehat{R}_{\mathrm{cPC}}(f)$ in Eq. (7) with the absolute value function; **Pcomp-Teacher**, which improves the RankPruning method by imposing consistency regularization to make the learning model consistent with a teacher model. Besides, we compare with the following baselines: **Binary-Biased**, which conducts binary classification by regarding the data from $\widetilde{p}_+(x)$ as positive data and the data from $\widetilde{p}_-(x)$ as negative data. This is a straightforward method to handle the Pcomp classification problem. In our setting, Binary-Biased reduces to the BER minimization method (Menon et al., 2015); **Noisy-Unbiased**, which is a noisy-label learning method that minimizes the empirical approximation of the URE proposed by Natarajan et al. (2013); **RankPruning**, which is a noisy-label learning method (Northcutt et al., 2017) that minimizes $\widehat{R}_{\mathrm{pPC}}(f)$ in Eq. (8). For all learning methods, we take the logistic loss as the binary loss function $\ell$ (i.e., $\ell(z) = \ln(1 + \exp(-z))$), for fair comparisons. We implement our methods using PyTorch (Paszke et al., 2019) and use the Adam (Kingma & Ba, 2015) optimization method with mini-batch size set to 256 and the number of training epochs set to 100. All the experiments are conducted on GeForce GTX 1080 Ti GPUs.

Table 2: Classification accuracy (mean±std) in percentage of each method on the four UCI datasets with different class priors. The best performance is highlighted in bold.

| Class Prior | Methods | USPS | Pendigits | Optdigits | CNAE-9 |
|---|---|---|---|---|---|
| $\pi_+ = 0.2$ | Noisy-Unbiased | 88.43±2.96 | 83.35±0.57 | 84.63±1.77 | 83.73±1.46 |
| | Binary-Biased | 79.37±1.86 | 65.24±5.48 | 65.23±3.48 | 63.48±1.87 |
| | RankPruning | 91.93±0.83 | 78.43±5.85 | 83.61±1.89 | 76.03±5.07 |
| | Pcomp-ABS | 90.94±0.83 | 86.14±0.72 | 85.98±1.82 | 82.40±1.42 |
| | Pcomp-ReLU | 91.90±0.60 | 86.35±0.80 | 87.55±1.35 | 82.97±1.26 |
| | Pcomp-Unbiased | 91.88±0.75 | 85.89±1.50 | **86.79±1.52** | **84.13±1.73** |
| | Pcomp-Teacher | **93.18±0.57** | **86.36±2.33** | 85.81±1.54 | 80.44±4.33 |
| $\pi_+ = 0.5$ | Noisy-Unbiased | 87.57±2.02 | 83.47±2.62 | 85.13±1.38 | 76.77±0.95 |
| | Binary-Biased | 90.78±0.44 | 79.60±5.46 | 81.84±3.98 | 74.34±1.41 |
| | RankPruning | 92.28±0.26 | 80.19±2.47 | 82.77±1.77 | 70.65±2.92 |
| | Pcomp-ABS | 89.81±1.29 | 83.32±2.38 | 83.61±1.78 | 76.32±1.38 |
| | Pcomp-ReLU | 91.10±0.73 | 84.26±2.37 | 84.43±1.52 | 76.58±1.17 |
| | Pcomp-Unbiased | 90.77±0.87 | **84.52±2.49** | **85.43±1.79** | **77.12±1.24** |
| | Pcomp-Teacher | **92.53±0.30** | 82.10±2.26 | 84.54±1.90 | 74.89±3.60 |
| $\pi_+ = 0.8$ | Noisy-Unbiased | 88.49±2.14 | 85.62±1.29 | 87.05±1.24 | 83.78±1.42 |
| | Binary-Biased | 72.94±1.36 | 63.63±4.36 | 68.83±2.70 | 60.45±0.95 |
| | RankPruning | 89.02±8.69 | 84.94±1.33 | 87.24±0.87 | 83.33±4.79 |
| | Pcomp-ABS | 90.96±0.84 | 89.20±2.70 | 88.93±1.12 | 82.72±1.76 |
| | Pcomp-ReLU | 92.09±1.53 | **89.59±2.57** | 89.13±0.67 | 83.97±1.05 |
| | Pcomp-Unbiased | 91.28±1.39 | 89.13±2.42 | 88.25±1.26 | **85.50±1.62** |
| | Pcomp-Teacher | **93.05±0.70** | 87.64±1.70 | **89.30±1.41** | 83.62±3.62 |

**Experimental Setup.** We test the performance of all learning methods under different class prior settings, i.e., $\pi_+$ is selected from $\{0.2, 0.5, 0.8\}$. It is worth noting that we could estimate $\pi_+$ according to our described data generation process. Specifically, we can exactly estimate $\widetilde{\pi}$ by counting the fraction of collected pairwise comparison data in all the sampled pairs of data. Since $\widetilde{\pi} = \pi_+^2 + \pi_- = \pi_+^2 + 1 - \pi_+$, we have $\pi_+ = 1/2 - \sqrt{\widetilde{\pi} - 3/4}$ (if $\pi_+ < \pi_-$) or $\pi_+ = 1/2 + \sqrt{\widetilde{\pi} - 3/4}$ (if $\pi_+ \geq \pi_-$). Therefore, if we know whether $\pi_+$ is larger than $\pi_-$, we could exactly estimate the true class prior $\pi_+$. For simplicity, we assume that the class prior $\pi_+$ is known for all the methods. We repeat the sampling-and-training process 5 times for all learning methods on all datasets and record the mean accuracy with standard deviation (mean±std).

**Experimental Results with Complex Models.** Table 1 records the classification performance of each method on the four benchmark datasets with different class priors. From Table 1, we have the following observations: 1) Binary-Biased always achieves the worst performance, which indicates that simply conducting binary classification cannot well solve our Pcomp classification problem; 2) Pcomp-Unbiased is is inferior to Pcomp-ABS and Pcomp-ReLU. This observation accords with what we have discussed, i.e., directly minimizing $\widehat{R}_{\mathrm{PC}}(f)$ would suffer from overfitting when complex models are used because there are negative terms included in $\widehat{R}_{\mathrm{PC}}(f)$ and the empirical risk can be negative during the training process. In contrast, Pcomp-ReLU and Pcomp-ABS employ consistent correction functions on $\widehat{R}_{\mathrm{PC}}(f)$ so that the empirical risk will never be negative. Therefore, when complex models such as deep neural networks are used, Pcomp-ReLU and Pcomp-ABS are expected to outperform Pcomp-Unbiased; 3) Pcomp-Teacher achieves the best performance in most cases. This observation verifies the effectiveness of the imposed consistency regularization, which makes the learning model consistent with a teacher model, for improving the quality of selected confident examples by the RankPruning method; 4) It is worth noting that the standard deviations of Binary-Biased, Pcomp-Unbiased, and Noisy-Unbiased are sometimes higher than other methods. This is because the three methods suffer from overfitting when complex models are used, and the performance could be quite unstable in different trials. In addition, Noisy-Unbiased holds the accuracy of 80.00±0.00% on CIFAR-10 with class prior 0.2. This extreme case happens because Noisy-Unbiased always simply classifies all examples into the negative class due to the serious overfitting issue on a complex class-imbalanced dataset with a complex model ResNet-34.

**Experimental Results with Simple Models.** Table 2 reports the classification performance of each method on the four UCI datasets with different class priors. From Table 2, we have the follow-

ing observations: 1) Binary-Biased achieves the worst performance in nearly all cases; 2) Pcomp-Unbiased is slightly better than Pcomp-ReLU and Pcomp-ABS, because Pcomp-Unbiased does not suffer from overfitting when the linear model is used, and it is not necessary to use consistent correction functions anymore. Besides, Pcomp-Unbiased becomes comparable to Pcomp-Teacher and achieves the best performance in half of the cases; 3) Pcomp-Teacher is still better than RankPruning, while it is sometimes inferior to Pcomp-Unbiased. This is because the linear model is not as powerful as neural networks, and the selected confident examples may not be so reliable.

## 6 CONCLUSION

In this paper, we proposed a novel weakly supervised learning setting called *pairwise comparison (Pcomp) classification*, where we aim to train a binary classifier from only *pairwise comparison data*, i.e., two examples that we know one is more likely to be positive than the other, instead of pointwise labeled data. Pcomp classification is useful for private classification tasks where we are not allowed to directly access labels and subjective classification tasks where labelers have different labeling standards. To solve the Pcomp classification problem, we presented a mathematical formulation for the generation process of pairwise comparison data, based on which we explored two *unbiased risk estimators* (UREs) to train a binary classifier by *empirical risk minimization* and established the corresponding *estimation error bounds*. We first proved that a URE can be derived and improved it using correction functions. Then, we started from the *noisy-label learning* perspective to introduce a *progressive* URE and improved it by imposing *consistency regularization*. Finally, experiments demonstrated the effectiveness of our proposed methods.

In future work, we will apply Pcomp classification to solve some challenging real-world problems like binary classification with class overlapping. In addition, we could also extend Pcomp classification to the multi-class classification setting by using the one-versus-all strategy. Suppose there are multiple classes, we are given pairs of unlabeled data that we know which one is more likely to belong to a specific class. Then, we can use the proposed methods in this paper to train a binary classifier for each class. Finally, by comparing the outputs of these binary classifiers, the predicted class can be determined.

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

# A    PROOF OF THEOREM 1

It is clear that each pair of examples $(\boldsymbol{x}, \boldsymbol{x}')$ is independently drawn from the following data distribution:

$$\widetilde{p}(\boldsymbol{x}, \boldsymbol{x}') = p((\boldsymbol{x}, \boldsymbol{x}') \mid (y, y') \in \widetilde{\mathcal{Y}}) = \frac{p((\boldsymbol{x}, \boldsymbol{x}'), (y, y') \in \widetilde{\mathcal{Y}})}{p((y, y') \in \widetilde{\mathcal{Y}})},$$

where $p((y, y') \in \widetilde{\mathcal{Y}}) = \pi_+^2 + \pi_-^2 + \pi_+\pi_-$ and

$$
\begin{aligned}
p(\boldsymbol{x}, \boldsymbol{x}', (y, y') \in \widetilde{\mathcal{Y}}) &= \sum_{(y,y') \in \widetilde{\mathcal{Y}}} p(\boldsymbol{x}, \boldsymbol{x}' \mid (y, y')) \cdot p(y, y') \\
&= \pi_+^2 p_+(\boldsymbol{x}) p_+(\boldsymbol{x}') + \pi_-^2 p_-(\boldsymbol{x}) p_-(\boldsymbol{x}') + \pi_+\pi_- p_+(\boldsymbol{x}) p_-(\boldsymbol{x}').
\end{aligned}
$$

Finally, let $\widetilde{p}(\boldsymbol{x}, \boldsymbol{x}') = p((\boldsymbol{x}, \boldsymbol{x}') \mid (y, y') \in \widetilde{\mathcal{Y}})$, the proof is completed. $\qquad\square$

# B    PROOF OF THEOREM 2

In order to decompose the pairwise comparison data distribution into pointwise distribution, we marginalize $\widetilde{p}(\boldsymbol{x}, \boldsymbol{x}')$ with respect to $\boldsymbol{x}$ or $\boldsymbol{x}'$. Then we can obtain

$$
\begin{aligned}
\int \widetilde{p}(\boldsymbol{x}, \boldsymbol{x}') \mathrm{d}\boldsymbol{x}' &= \frac{1}{\widetilde{\pi}} \left( \pi_+^2 p_+(\boldsymbol{x}) + \pi_-^2 p_-(\boldsymbol{x}) + \pi_+\pi_- p_+(\boldsymbol{x}) \right) \\
&= \frac{\pi_+}{\pi_-^2 + \pi_+} p_+(\boldsymbol{x}) + \frac{\pi_-^2}{\pi_-^2 + \pi_+} p_-(\boldsymbol{x}) \\
&= \widetilde{p}_+(\boldsymbol{x}),
\end{aligned}
$$

and

$$
\begin{aligned}
\int \widetilde{p}(\boldsymbol{x}, \boldsymbol{x}') \mathrm{d}\boldsymbol{x} &= \frac{1}{\widetilde{\pi}} \left( \pi_+^2 p_+(\boldsymbol{x}') + \pi_-^2 p_-(\boldsymbol{x}') + \pi_+\pi_- p_-(\boldsymbol{x}') \right) \\
&= \frac{\pi_+^2}{\pi_+^2 + \pi_-} p_+(\boldsymbol{x}') + \frac{\pi_-}{\pi_+^2 + \pi_-} p_-(\boldsymbol{x}') \\
&= \widetilde{p}_-(\boldsymbol{x}'),
\end{aligned}
$$

which concludes the proof of Theorem 2. $\qquad\square$

# C    PROOF OF LEMMA 1

Based on Theorem 2, we can obtain the following linear equation:

$$
\begin{bmatrix} \widetilde{p}_+(\boldsymbol{x}) \\ \widetilde{p}_-(\boldsymbol{x}) \end{bmatrix} = \frac{1}{\widetilde{\pi}} \begin{bmatrix} \pi_+ & \pi_-^2 \\ \pi_+^2 & \pi_- \end{bmatrix} \begin{bmatrix} p_+(\boldsymbol{x}) \\ p_-(\boldsymbol{x}) \end{bmatrix}.
$$

By solving the above equation, we obtain

$$
p_+(\boldsymbol{x}) = \frac{1}{\pi_+ - \pi_-\pi_+^2} \left( \widetilde{\pi} \cdot \widetilde{p}_+(\boldsymbol{x}) - \pi_-\widetilde{\pi} \cdot \widetilde{p}_-(\boldsymbol{x}) \right) = \frac{1}{\pi_+} \left( \widetilde{p}_+(\boldsymbol{x}) - \pi_-\widetilde{p}_-(\boldsymbol{x}) \right),
$$

$$
p_-(\boldsymbol{x}) = \frac{1}{\pi_- - \pi_+\pi_-^2} \left( \widetilde{\pi} \cdot \widetilde{p}_-(\boldsymbol{x}) - \pi_+\widetilde{\pi} \cdot \widetilde{p}_+(\boldsymbol{x}) \right) = \frac{1}{\pi_-} \left( \widetilde{p}_-(\boldsymbol{x}) - \pi_+\widetilde{p}_+(\boldsymbol{x}) \right),
$$

which concludes the proof of Lemma 1. $\qquad\square$

## D   PROOF OF THEOREM 3

It is quite intuitive to derive

$$
\begin{aligned}
R(f) &= \mathbb{E}_{p(\boldsymbol{x},y)}\big[\ell(f(\boldsymbol{x}),y)\big] \\
&= \pi_+ \mathbb{E}_{p_+(\boldsymbol{x})}\big[\ell(f(\boldsymbol{x}),+1)\big] + \pi_- \mathbb{E}_{p_-(\boldsymbol{x})}\big[\ell(f(\boldsymbol{x}),-1)\big] \\
&= \frac{\pi_+\widetilde{\pi}}{\pi_+ - \pi_-\pi_+^2} \mathbb{E}_{\widetilde{p}_+(\boldsymbol{x})}\big[\ell(f(\boldsymbol{x}),+1)\big] - \frac{\pi_+\pi_-\widetilde{\pi}}{\pi_+ - \pi_-\pi_+^2} \mathbb{E}_{\widetilde{p}_-(\boldsymbol{x'})}\big[\ell(f(\boldsymbol{x}),+1)\big] \qquad \text{(Lemma 1)} \\
&\qquad + \frac{\pi_-\widetilde{\pi}}{\pi_- - \pi_+\pi_-^2} \mathbb{E}_{\widetilde{p}_-(\boldsymbol{x'})}\big[\ell(f(\boldsymbol{x}),-1)\big] - \frac{\pi_+\pi_-\widetilde{\pi}}{\pi_- - \pi_+\pi_-^2} \mathbb{E}_{\widetilde{p}_+(\boldsymbol{x})}\big[\ell(f(\boldsymbol{x}),-1)\big] \\
&= \mathbb{E}_{\widetilde{p}_+(\boldsymbol{x})}\big[\ell(f(\boldsymbol{x}),+1) - \pi_+\ell(f(\boldsymbol{x}),-1)\big] + \mathbb{E}_{\widetilde{p}_-(\boldsymbol{x'})}\big[\ell(f(\boldsymbol{x}),-1) - \pi_-\ell(f(\boldsymbol{x}),+1)\big] \\
&= R_{\mathrm{PC}}(f),
\end{aligned}
$$

which concludes the proof of Theorem 3. $\qquad\qquad\square$

## E   PROOF OF THEOREM 4

First of all, we introduce the following notations:

$$
\begin{aligned}
R_{\mathrm{PC}}^+(f) &= \mathbb{E}_{\widetilde{p}_+(\boldsymbol{x})}\big[\ell(f(\boldsymbol{x}),+1) - \pi_+\ell(f(\boldsymbol{x}),-1)\big], \\
\widehat{R}_{\mathrm{PC}}^+(f) &= \frac{1}{n}\sum_{i=1}^{n}\Big(\ell(f(\boldsymbol{x}_i),+1) - \pi_+\ell(f(\boldsymbol{x}_i),-1)\Big), \\
R_{\mathrm{PC}}^-(f) &= \mathbb{E}_{\widetilde{p}_-(\boldsymbol{x'})}\big[\ell(f(\boldsymbol{x'}),-1) - \pi_-\ell(f(\boldsymbol{x'}),+1)\big], \\
\widehat{R}_{\mathrm{PC}}^-(f) &= \frac{1}{n}\sum_{i=1}^{n}\Big(\ell(f(\boldsymbol{x'}_i),-1) - \pi_-\ell(f(\boldsymbol{x'}_i),+1)\Big).
\end{aligned}
$$

In this way, we could simply represent $R_{\mathrm{PC}}(f)$ and $\widehat{R}_{\mathrm{PC}}(f)$ as

$$
R_{\mathrm{PC}}(f) = R_{\mathrm{PC}}^+(f) + R_{\mathrm{PC}}^-(f), \quad \widehat{R}_{\mathrm{PC}}(f) = \widehat{R}_{\mathrm{PC}}^+(f) + \widehat{R}_{\mathrm{PC}}^-(f).
$$

Then we have the following lemma.

**Lemma 2.** *The following inequality holds:*

$$
R(\widehat{f}_{\mathrm{PC}}) - R(f^\star) \leq 2\sup_{f\in\mathcal{F}}\Big|R_{\mathrm{PC}}^+(f) - \widehat{R}_{\mathrm{PC}}^+(f)\Big| + 2\sup_{f\in\mathcal{F}}\Big|R_{\mathrm{PC}}^-(f) - \widehat{R}_{\mathrm{PC}}^-(f)\Big|. \qquad (9)
$$

*Proof.* We could intuitively express $R(\widehat{f}_{\mathrm{PC}}) - R(f^\star)$ as

$$
\begin{aligned}
R(\widehat{f}_{\mathrm{PC}}) - R(f^\star) &= R(\widehat{f}_{\mathrm{PC}}) - \widehat{R}_{\mathrm{PC}}(\widehat{f}_{\mathrm{PC}}) + \widehat{R}_{\mathrm{PC}}(\widehat{f}_{\mathrm{PC}}) - \widehat{R}_{\mathrm{PC}}(f^\star) + \widehat{R}_{\mathrm{PC}}(f^\star) - R(f^\star) \\
&= R_{\mathrm{PC}}(\widehat{f}_{\mathrm{PC}}) - \widehat{R}_{\mathrm{PC}}(\widehat{f}_{\mathrm{PC}}) + \widehat{R}_{\mathrm{PC}}(\widehat{f}_{\mathrm{PC}}) - \widehat{R}_{\mathrm{PC}}(f^\star) + \widehat{R}_{\mathrm{PC}}(f^\star) - R_{\mathrm{PC}}(f^\star) \\
&\leq \sup_{f\in\mathcal{F}}\Big|R_{\mathrm{PC}}(f) - \widehat{R}_{\mathrm{PC}}(f)\Big| + 0 + \sup_{f\in\mathcal{F}}\Big|R_{\mathrm{PC}}(f) - \widehat{R}_{\mathrm{PC}}(f)\Big| \\
&= 2\sup_{f\in\mathcal{F}}\Big|R_{\mathrm{PC}}(f) - \widehat{R}_{\mathrm{PC}}(f)\Big| \\
&\leq 2\sup_{f\in\mathcal{F}}\Big|R_{\mathrm{PC}}^+(f) - \widehat{R}_{\mathrm{PC}}^+(f)\Big| + 2\sup_{f\in\mathcal{F}}\Big|R_{\mathrm{PC}}^-(f) - \widehat{R}_{\mathrm{PC}}^-(f)\Big|,
\end{aligned}
$$

where the second inequality holds due to Theorem 3. $\qquad\square$

As suggested by Lemma 2, we need to further upper bound the right hand size of Eq. (9). Before doing that, we introduce the *uniform deviation bound*, which is useful to derive estimation error bounds. The proof can be found in some textbooks such as Mohri et al. (2012) (Theorem 3.1).

**Lemma 3.** *Let $Z$ be a random variable drawn from a probability distribution with density $\mu$, $\mathcal{H} = \{h : \mathcal{Z} \mapsto [0, M]\}$ ($M > 0$) be a class of measurable functions, $\{z_i\}_{i=1}^n$ be i.i.d. examples drawn from the distribution with density $\mu$. Then, for any $delta > 0$, with probability at least $1 - \delta$,*

$$\sup_{h \in \mathcal{H}} \left| \mathbb{E}_{Z \sim \mu}\big[h(Z)\big] - \frac{1}{n} \sum_{i=1}^n h(z_i) \right| \leq 2\mathfrak{R}_n(\mathcal{H}) + M\sqrt{\frac{\log \frac{2}{\delta}}{2n}},$$

*where $\mathfrak{R}_n(\mathcal{H})$ denotes the (expected)* Rademacher complexity *([Bartlett & Mendelson, 2002](#)) of $\mathcal{H}$ with sample size $n$ over $\mu$.*

**Lemma 4.** *Suppose the loss function $\ell$ is $\rho$-Lipschitz with respect to the first argument ($0 < \rho < \infty$), and all the functions in the model class $\mathcal{F}$ are bounded, i.e., there exists a constant $C_{\mathrm{b}}$ such that $\|f\|_\infty \leq C_{\mathrm{b}}$ for any $f \in \mathcal{F}$. Let $C_\ell := \sup_{t=\pm 1} \ell(C_{\mathrm{b}}, t)$. For any $\delta > 0$, with probability $1 - \delta$,*

$$\sup_{f \in \mathcal{F}} \left| R_{\mathrm{PC}}^+(f) - \widehat{R}_{\mathrm{PC}}^+(f) \right| \leq (1 + \pi_+)2\rho\widetilde{\mathfrak{R}}_n^+(\mathcal{F}) + (1 + \pi_+)C_\ell\sqrt{\frac{\log \frac{4}{\delta}}{2n}}.$$

*Proof.* By the definition of $R_{\mathrm{PC}}^+(f)$ and $\widehat{R}_{\mathrm{PC}}^+(f)$, we can obtain

$$\sup_{f \in \mathcal{F}} \left| R_{\mathrm{PC}}^+(f) - \widehat{R}_{\mathrm{PC}}^+(f) \right| \leq \sup_{f \in \mathcal{F}} \left| \mathbb{E}_{\widetilde{p}_+(\boldsymbol{x})}\big[\ell(f(\boldsymbol{x}), +1)\big] - \frac{1}{n} \sum_{i=1}^n \ell(f(\boldsymbol{x}), +1) \right|$$

$$+ \pi_+ \sup_{f \in \mathcal{F}} \left| \mathbb{E}_{\widetilde{p}_+(\boldsymbol{x})}\big[\ell(f(\boldsymbol{x}), -1)\big] - \frac{1}{n} \sum_{i=1}^n \ell(f(\boldsymbol{x}), -1) \right|. \tag{10}$$

By applying Lemma 3, we have for any $\delta > 0$, with probability $1 - \delta$,

$$\sup_{f \in \mathcal{F}} \left| \mathbb{E}_{\widetilde{p}_+(\boldsymbol{x})}\big[\ell(f(\boldsymbol{x}), +1)\big] - \frac{1}{n} \sum_{i=1}^n \ell(f(\boldsymbol{x}), +1) \right| \leq 2\widetilde{\mathfrak{R}}_n^+(\ell \circ \mathcal{F}) + C_\ell\sqrt{\frac{\log \frac{2}{\delta}}{2n}}, \tag{11}$$

and for any for any $\delta > 0$, with probability $1 - \delta$,

$$\sup_{f \in \mathcal{F}} \left| \mathbb{E}_{\widetilde{p}_+(\boldsymbol{x})}\big[\ell(f(\boldsymbol{x}), -1)\big] - \frac{1}{n} \sum_{i=1}^n \ell(f(\boldsymbol{x}), -1) \right| \leq 2\widetilde{\mathfrak{R}}_n^+(\ell \circ \mathcal{F}) + C_\ell\sqrt{\frac{\log \frac{2}{\delta}}{2n}}, \tag{12}$$

where $\ell \circ \mathcal{F}$ means $\{\ell \circ f \mid f \in \mathcal{F}\}$. By Talagrand's lemma (Lemma 4.2 in [Mohri et al. (2012)](#)),

$$\widetilde{\mathfrak{R}}_n^+(\ell \circ \mathcal{F}) \leq \rho\widetilde{\mathfrak{R}}_n^+(\mathcal{F}). \tag{13}$$

Finally, by combing Eqs. (10), (11), (12), and (13), we have for any $\delta > 0$, with probability at least $1 - \delta$,

$$\sup_{f \in \mathcal{F}} \left| R_{\mathrm{PC}}^+(f) - \widehat{R}_{\mathrm{PC}}^+(f) \right| \leq (1 + \pi_+)2\rho\widetilde{\mathfrak{R}}_n^+(\mathcal{F}) + (1 + \pi_+)C_\ell\sqrt{\frac{\log \frac{4}{\delta}}{2n}}, \tag{14}$$

which concludes the proof of Lemma 4. $\qquad\square$

**Lemma 5.** *Suppose the loss function $\ell$ is $\rho$-Lipschitz with respect to the first argument ($0 < \rho < \infty$), and all the functions in the model class $\mathcal{F}$ are bounded, i.e., there exists a constant $C_{\mathrm{b}}$ such that $\|f\|_\infty \leq C_{\mathrm{b}}$ for any $f \in \mathcal{F}$. Let $C_\ell := \sup_{t=\pm 1} \ell(C_{\mathrm{b}}, t)$. For any $\delta > 0$, with probability $1 - \delta$,*

$$\sup_{f \in \mathcal{F}} \left| R_{\mathrm{PC}}^-(f) - \widehat{R}_{\mathrm{PC}}^-(f) \right| \leq (1 + \pi_-)2\rho\widetilde{\mathfrak{R}}_n^-(\mathcal{F}) + (1 + \pi_-)C_\ell\sqrt{\frac{\log \frac{4}{\delta}}{2n}}.$$

*Proof.* Lemma 5 can be proved similarly to Lemma 4. $\qquad\square$

By combining Lemma 2, Lemma 4, and Lemma 5, Theorem 4 is proved. $\qquad\square$

# F   PROOF OF THEOREM 5

Suppose there are $n$ pairs of paired data points, which means there are in total $2n$ data points. For our Pcomp classification problem, we could simply regard $\boldsymbol{x}$ sampled from $\widetilde{p}_+(\boldsymbol{x})$ as (noisy) positive data and $\boldsymbol{x}'$ sampled from $\widetilde{p}_-(\boldsymbol{x}')$ as (noisy) negative data. Given $n$ pairs of examples $\{(\boldsymbol{x}_i, \boldsymbol{x}'_i)\}_{i=1}^n$, for the $n$ observed positive examples, there are actually $n \cdot p(y = +1 | \widetilde{y} = +1)$ true positive examples; for the $n$ observed negative examples, there are actually $n \cdot p(y = -1 | \widetilde{y} = -1)$ true negative examples. From our defined data generation process in Theorem 1, it is intuitive to obtain

$$p(y = +1 \mid \widetilde{y} = +1) = \frac{\pi_+^2 + \pi_+\pi_-}{\pi_+^2 + \pi_-^2 + \pi_+\pi_-},$$

$$p(y = -1 \mid \widetilde{y} = -1) = \frac{\pi_-^2 + \pi_+\pi_-}{\pi_+^2 + \pi_-^2 + \pi_+\pi_-}.$$

Since $\phi_+ = p(y = -1 \mid \widetilde{y} = +1) = 1 - p(y = +1 \mid \widetilde{y} = +1)$ and $\phi_- = p(y = +1 \mid \widetilde{y} = -1) = 1 - p(y = -1 \mid \widetilde{y} = -1)$, we can obtain

$$\phi_+ = p(y = -1 \mid \widetilde{y} = +1) = 1 - \frac{\pi_+^2 + \pi_+\pi_-}{\pi_+^2 + \pi_-^2 + \pi_+\pi_-} = \frac{\pi_-^2}{\pi_+^2 + \pi_-^2 + \pi_+\pi_-},$$

$$\phi_- = p(y = +1 \mid \widetilde{y} = -1) = 1 - \frac{\pi_-^2 + \pi_+\pi_-}{\pi_+^2 + \pi_-^2 + \pi_+\pi_-} = \frac{\pi_+^2}{\pi_+^2 + \pi_-^2 + \pi_+\pi_-}.$$

In this way, we can further obtain the following noise transition ratios:

$$\rho_+ = p(\widetilde{y} = -1 \mid y = +1) = \frac{p(y = +1 \mid \widetilde{y} = -1)p(\widetilde{y} = -1)}{p(y = +1)} = \frac{\pi_+}{2(\pi_+^2 + \pi_-^2 + \pi_+\pi_-)},$$

$$\rho_- = p(\widetilde{y} = +1 \mid y = -1) = \frac{p(y = -1 \mid \widetilde{y} = +1)p(\widetilde{y} = +1)}{p(y = -1)} = \frac{\pi_-}{2(\pi_+^2 + \pi_-^2 + \pi_+\pi_-)},$$

where $p(\widetilde{y} = 1) = p(\widetilde{y} = -1) = \frac{1}{2}$, because we have the same number of observed positive examples and negative examples.

# G   PROOF OF THEOREM 7

First of all, we introduce the following notations:

$$R_{\text{pPC}}^+(f) = \mathbb{E}_{\widetilde{p}_+(\boldsymbol{x})}\big[\ell(f(\boldsymbol{x}), +1)\mathbb{I}[\boldsymbol{x} \in P\widetilde{P}]\big],$$

$$\widehat{R}_{\text{pPC}}^+(f) = \frac{1}{n} \sum_{i=1}^n \big(\ell(f(\boldsymbol{x}_i), +1)\mathbb{I}[\boldsymbol{x}_i \in P\widetilde{P}]\big),$$

$$R_{\text{pPC}}^-(f) = \mathbb{E}_{\widetilde{p}_-(\boldsymbol{x}')}\big[\ell(f(\boldsymbol{x}'), -1)\mathbb{I}[\boldsymbol{x}' \in N\widetilde{N}]\big],$$

$$\widehat{R}_{\text{pPC}}^-(f) = \frac{1}{n} \sum_{i=1}^n \big(\ell(f(\boldsymbol{x}'_i), -1)\mathbb{I}[\boldsymbol{x}'_i \in N\widetilde{N}]\big).$$

In this way, we could simply represent $R_{\text{ppc}}(f)$ and $\widehat{R}_{\text{pPC}}(f)$ as

$$R_{\text{pPC}}(f) = \frac{1}{1 - \rho_+}R_{\text{pPC}}^+(f) + \frac{1}{1 - \rho_-}R_{\text{pPC}}^-(f), \quad \widehat{R}_{\text{pPC}}(f) = \frac{1}{1 - \rho_+}\widehat{R}_{\text{pPC}}^+(f) + \frac{1}{1 - \rho_-}\widehat{R}_{\text{pPC}}^-(f).$$

Then we have the following lemma.

**Lemma 6.** *The following inequality holds:*

$$R(\widehat{f}_{\text{pPC}}) - R(f^\star) \leq \frac{2}{1 - \rho_+} \sup_{f \in \mathcal{F}} \left| R_{\text{pPC}}^+(f) - \widehat{R}_{\text{pPC}}^+(f) \right| + \frac{2}{1 - \rho_-} \sup_{f \in \mathcal{F}} \left| R_{\text{pPC}}^-(f) - \widehat{R}_{\text{pPC}}^-(f) \right|.$$

$$(15)$$

*Proof.* We omit the proof of Lemma 6 since it is quite similar to that of Lemma 2. □

As suggested by Lemma 6, we need to further upper bound the right hand size of Eq. (15). According to Lemma 3, we have the following two lemmas.

**Lemma 7.** *Suppose the loss function $\ell$ is $\rho$-Lipschitz with respect to the first argument ($0 < \rho < \infty$), and all the functions in the model class $\mathcal{F}$ are bounded, i.e., there exists a constant $C_{\mathrm{b}}$ such that $\|f\|_\infty \le C_{\mathrm{b}}$ for any $f \in \mathcal{F}$. Let $C_\ell := \sup_{z \le C_{\mathrm{b}}, t = \pm 1} \ell(z, t)$. For any $\delta > 0$, with probability $1 - \delta$,*

$$\sup_{f \in \mathcal{F}} \left| R_{\mathrm{pPC}}^+(f) - \widehat{R}_{\mathrm{pPC}}^+(f) \right| \le 2\rho \widetilde{\mathfrak{R}}_n^+(\mathcal{F}) + C_\ell \sqrt{\frac{\log \frac{2}{\delta}}{2n}}.$$

**Lemma 8.** *Suppose the loss function $\ell$ is $\rho$-Lipschitz with respect to the first argument ($0 < \rho < \infty$), and all the functions in the model class $\mathcal{F}$ are bounded, i.e., there exists a constant $C_{\mathrm{b}}$ such that $\|f\|_\infty \le C_{\mathrm{b}}$ for any $f \in \mathcal{F}$. Let $C_\ell := \sup_{z \le C_{\mathrm{b}}, t = \pm 1} \ell(z, t)$. For any $\delta > 0$, with probability $1 - \delta$,*

$$\sup_{f \in \mathcal{F}} \left| R_{\mathrm{pPC}}^-(f) - \widehat{R}_{\mathrm{pPC}}^-(f) \right| \le 2\rho \widetilde{\mathfrak{R}}_n^-(\mathcal{F}) + C_\ell \sqrt{\frac{\log \frac{2}{\delta}}{2n}}.$$

We omit the proofs of Lemma 7 and Lemma 8 since they are similar to that of Lemma 4.

By combing Lemma 6, Lemma 7, and Lemma 8, Theorem 7 is proved.

## H  SUPPLEMENTARY INFORMATION OF EXPERIMENTS

Table 3 reports the specification of the used benchmark datasets and models.

**MNIST**[2] (LeCun et al., 1998).   This is a grayscale image dataset composed of handwritten digits from 0 to 9 where the size of the each image is $28 \times 28$. It contains 60,000 training images and 10,000 test images. Because the original dataset has 10 classes, we regard the even digits as the positive class and the odd digits as the negative class.

**Fashion-MNIST**[3] (Xiao et al., 2017).   Similarly to MNIST, this is also a grayscale image dataset composed of fashion items ('T-shirt', 'trouser', 'pullover', 'dress', 'sandal', 'coat', 'shirt', 'sneaker', 'bag', and 'ankle boot'). It contains 60,000 training examples and 10,000 test examples. It is converted into a binary classification dataset as follows:

- The positive class is formed by 'T-shirt', 'pullover', 'coat', 'shirt', and 'bag'.
- The negative class is formed by 'trouser', 'dress', 'sandal', 'sneaker', and 'ankle boot'.

**Kuzushiji-MNIST**[4] (Netzer et al., 2011).   This is another grayscale image dataset that is similar to MNIST. It is a 10-class dataset of cursive Japanese ("Kuzushiji") characters. It consists of 60,000 training images and 10,000 test images. It is converted into a binary classification dataset as follows:

- The positive class is formed by 'o', 'su','na', 'ma', 're'.
- The negative class is formed by 'ki','tsu','ha', 'ya','wo'.

**CIFAR-10**[5] (Krizhevsky et al., 2009).   This is also a color image dataset of 10 different objects ('airplane', 'bird', 'automobile', 'cat', 'deer', 'dog', 'frog', 'horse', 'ship', and 'truck'), where the size of each image is $32 \times 32 \times 3$. There are 5,000 training images and 1,000 test images per class. This dataset is converted into a binary classification dataset as follows:

- The positive class is formed by 'bird', 'deer', 'dog', 'frog', 'cat', and 'horse'.
- The negative class is formed by 'airplane', 'automobile', 'ship', and 'truck'.

---

[2] http://yann.lecun.com/exdb/mnist/
[3] https://github.com/zalandoresearch/fashion-mnist
[4] https://github.com/rois-codh/kmnist
[5] https://www.cs.toronto.edu/~kriz/cifar.html

Table 3: Specification of the used benchmark datasets and models.

| Dataset | # Train | # Test | # Features | # Classes | Model |
|---|---|---|---|---|---|
| MNIST | 60,000 | 10,000 | 784 | 10 | MLP ($d$-300-300-300-300-1) |
| Fashion-MNIST | 60,000 | 10,000 | 784 | 10 | MLP ($d$-300-300-300-300-1) |
| Kuzushiji-MNIST | 60,000 | 10,000 | 784 | 10 | MLP ($d$-300-300-300-300-1) |
| CIFAR-10 | 50,000 | 10,000 | 3,072 | 10 | ResNet-34 |
| USPS | 7,437 | 1,861 | 256 | 10 | Linear Model ($d$-1) |
| Pendigits | 8,793 | 2,199 | 16 | 10 | Linear Model ($d$-1) |
| Optdigits | 4,495 | 1,125 | 62 | 10 | Linear Model ($d$-1) |
| CNAE-9 | 864 | 216 | 856 | 9 | Linear Model ($d$-1) |

**USPS, Pendigits, Optdigits.** These datasets are composed of handwritten digits from 0 to 9. Because each of the original datasets has 10 classes, we regard the even digits as the positive class and the odd digits as the negative class.

**CNAE-9.** This dataset contains 1,080 documents of free text business descriptions of Brazilian companies categorized into a subset of 9 categories cataloged in a table called National Classification of Economic Activities.

- The positive class is formed by '2', '4', '6' and '8'.
- The negative class is formed by '1', '3', '5', '7' and '9'.

For MNIST, Kuzushiji-MNIST, and Fashion-MNIST, we set learning rate to $1e-3$ and weight decay to $1e-5$. For CIFAR-10, we set learning rate to $1e-3$ and weight decay to $1e-3$. We also list the number of pointwise corrupted examples used for model training on each dataset: 30,000 for MNIST, Kuzushiji-MNIST, Fashion-MNIST, and CIFAR-10; 4,000 for USPS; 5,000 for Pendigits; 2,000 for Optdigits; 400 for CNAE-9.

