# OpenReview forum: "Pointwise Binary Classification with Pairwise Confidence Comparisons"
_ICLR.cc/2021/Conference — Reject_

### Official Review · AnonReviewer1 · 2020-10-19
**Is the proposed method good enough?**

**Rating:** 5
**Confidence:** 4

**Review:**

Summary of the paper:
The paper studies binary classification with pairwise comparisons as the supervision. Instead of the traditional pointwise supervision, the paper assumes only access to pairs of examples $x,x'$ where we believe that $p(y=1|x)\geq p(y=1|x')$. The authors considers two kinds of methods. The first one considers x, x' be sampled from two different distributions, and uses the method from Unlabeled-Unlabeled (UU) classification to obtain an unbiased risk estimator. The second one is the RankPruning method, where a fraction of data with highest confidence is selected and is used to derive a URE on the selected samples. The paper proposes to add a moving-average regularization to RankPruning. The authors performed experiments on 2-class versions of standard benchmark datasets. In general RankPruning with regularization performs the best in most cases, but the first method is also competitive.

Review:
The paper studies an important problem of learning classification from pairwise comparisons. I have some doubts though:
	a) The paper assumes a generative process of the pairs: Generally, it assumes a rejecting sampling process. If we sample $(x,x')$ with $y=-1,y'=+1$, then we reject it; otherwise we keep it. Why is this correct? Do we have data supporting this process? I may suggest other models: E.g., return $(x',x)$ if $y=-1,y'=+1$, and $(x,x')$ otherwise. Is this also plausible? How can we test what generation method is used?
	b) Another bigger problem is that the method ignores the PAIR nature of the problem; the method basically ignores the pairs and just treat list of $x, x'$ as from different distributions. We naturally loses the information that $p(y=1|x)\geq p(y=1|x')$. Is this a good method? Can we use other methods to utilize the pair information, such as siamese networks? How does the performance compare?
	c) In general the paper lacks a bit novelty - the first method is an adaptation of UU estimator, and the second method is just adding a regularization to RankPruning.

Minor Comments:
	a) Second paragraph in Sec 3 - the paragraph is a bit unclear to me. Can you state the generation process formally?
	b) Classification with pairwise comparisons is also considered in these works:
	Xu, Y., Zhang, H., Miller, K., Singh, A., and Dubrawski, A. Noise-tolerant interactive learning from pairwise comparisons with near-minimal label complexity. NIPS2017.
	D. M. Kane, S. Lovett, S. Moran, and J. Zhang. Active classification with comparison queries. FOCS 2017.
Both papers use a small amount of labeled data; but they considers the pair information.

---

> ### Author Response · Authors · 2020-11-18
> **Response to Reviewer #1**
>
> Thank you for your valuable comments!
>
> Q1. The assumption on the generative process of pairwise comparison examples.
>
> A1. Thanks for raising this concern. We adopt this assumption because it is harder to collect the positive confidence $p(y=+1|x)$ than the label $y$ for each example $(x,y)$. Our assumption is reasonable, as we generate pairwise comparison examples based on the labels rather than positive confidences, which is also supported by the third paragraph in Introduction. Besides, we can obtain an expression of the probability density of pairwise comparison examples, based on which we can derive unbiased risk estimators, so that the learning consistency can be guaranteed.
>
> It is also worth noting that a pair of $(x, x')$ with labels $(-1, +1)$ would not be discarded. Specifically, for a pair of data $(x, x')$ with labels $(-1, +1)$, $(x', x)$ could be taken as a pairwise comparison example according to our assumed generation process. Because by exchanging the positions of $(x, x')$, $(x', x)$ would be associated with labels $(+1, -1)$, which belong to the three cases $\{(+1, -1), (+1, -1), (-1, -1)\}$. Thank you for providing us another generation model, while your suggested generation model is exactly the same as ours, because we could take $(x', x)$ as a pairwise comparison example, given a pair of data $(x, x')$ with labels $(-1, +1)$.
>
> Q2. How can we test what generation method is used?
>
> A2. Thanks for the interesting question. It would be quite hard to test what generation method is used by the collected pairwise comparison data. Fortunately, as we showed that if the generation process of pairwise comparisons could be transformed into the generation process of noisy labels, we may estimate the noise rates by using some previous methods such as:
>
> Classification with noisy labels by importance reweighting. IEEE TPAMI 2016.
>
> An efficient and provable approach for mixture proportion estimation using linear independence assumption. CVPR 2018.
>
> Q3. Another bigger problem is that the method ignores the PAIR nature of the problem.
>
> A3. Thanks for raising this concern. We agree with you that we treat $x$ and $x'$ as data from different distributions, while we do not ignore the pair nature of the Pcomp classification problem. Specifically, for each pairwise comparison example $(x, x')$ (which means $x$ is more likely to be positive than $x'$), we aim to increase the prediction score of $x$ and decrease that of $x'$ (see Eq. (6)). This accords with the pair information that $x$ should have a larger positive confidence than $x'$. We might also use other methods to utilize the pair information (e.g., learning to rank and siamese networks), however, these methods may not be directly used to conduct effective binary classification even if the pair information is fully exploited. For example, even if there exists an oracle that can always provide perfect pairwise comparison information of any two examples, it still cannot figure out the latent labels of the two examples.
>
> Q4. Methods lack a bit of novelty.
>
> A4. Thank you for raising this concern. We admit the closeness between our first method and the UU estimator. However, we did not simply adapt the UU estimator to our method. Instead, we first assumed a data generation model (Theorem 1 and Theorem 2), and then derived an unbiased risk estimator (Theorem 3), showing its relation to the UU estimator ( Corollary 1). Such a whole derivation process is vitally important because we tried to solve this new Pcomp classification problem from scratch, while we eventually found that our derived method is closely related to the UU estimator. It is worth noting that the UU estimator is very general for weakly supervised binary classification, including:
>
> Analysis of learning from positive and unlabeled data. NeurIPS 2014.
>
> Convex formulation for learning from positive and unlabeled data. ICML 2015.
>
> Positive-unlabeled learning with non-negative risk estimator. NeurIPS 2017.
>
> Classification from pairwise similarity and unlabeled data. ICML 2018.
>
> Showing the relationships between our method and the UU estimator might be taken as a little contribution, which further extends the application scope of the UU estimator.
>
> As the first proposed method may not perform well when deep neural networks are used, we started from the noisy-label learning perspective to solve the Pcomp classification problem and proposed the second method, which improves RankPruning by adding a consistency regularization. Therefore, we aimed to provide effective solutions for thoroughly solving the Pcomp classification problem when different models are used.
>
> Q5. The second paragraph in Sec 3 is a bit unclear.
>
> A5. Thanks for pointing out this issue. We have carefully revised this part to make it clearer.
>
> Q6. Classification with pairwise comparisons is also considered in previous works.
>
> A6. Thanks for pointing out the two related papers. We have discussed the two papers in Section 2.

---

> ### Comment · AnonReviewer1 · 2020-11-25
> **Thanks for the response**
>
> I've read the author's response.
>
> Q1. Thanks for the revision - from the previous version it is really hard to interpret whether you reject labels (-1,+1) or you reverse the order. However, I don't think my question is resolved: What is the model IS rejecting (-1, +1)? Like keeping all the three combinations of labels but throwing away (-1, +1). Nevertheless, the model needs a justification.
>
> Q3. Correct me if I miss anything - Equation (6) can break into several sums over $x_i$ and $x_i'$ separately. Isn't this summing over all $x,x'$ and ignoring the pair nature? Maybe this is OK but I still feel it is worth noting.

---

> > ### Author Response · Authors · 2020-11-25
> > **Thanks for the reply**
> >
> > A1. Thank you for this question.
> >
> > The generation model works in the following way:
> >
> > For a pair of data $(x,x')$ with labels $(+1,+1)$ or $(+1,-1)$ or $(-1,-1)$, $(x,x')$ will be taken as a pairwise comparison example for Pcomp classification.
> >
> > For a pair of data $(x,x')$ with labels $(-1,+1)$, $(x',x)$ (order reversed) will be taken as a pairwise comparison example  for Pcomp classification.
> >
> > Therefore, the generation model will not throw away any data.
> >
> > We have provided more justifications for the generation model in the second and the third paragraph in Section 3. Please check that in our updated manuscript.
> >
> > A3. Thank you for this concern.
> >
> > Yes, Eq. (6) can break into several sums over $x_i$ and $x_i^\prime$ separately. However, the pair nature is not ignored. The pair nature in our problem setting actually means keeping the positive confidence comparison between two examples, instead of always processing two examples as an integrated pair. The key point is that for each pairwise comparison example $(x,x')$, $x$ should have a larger positive confidence than $x'$. Our methods have achieved this goal.

---

### Official Review · AnonReviewer4 · 2020-10-30
**Well motivated problem formulation with theoretical performance guarantees and extensive simulation results -- vote for accept.**

**Rating:** 7
**Confidence:** 3

**Review:**

Authors addressed the problem of 'Pcomp', a weakly supervised binary classification setting where the dataset only includes pairs of unlabeled data with an indicator of which one is more likely to be positive than the other (unlike pointwise labeled data in classical binary classification). The setting could be useful in privacy-preserving applications.

Authors proposed a generative model for pairwise data comparison, for which unbiased risk estimators could be obtained, and this crucially helps them to obtain a theoretical bound on the empirical risk minimizer.

I have a slight concern about the practicability of the assumed comparison model (Sec 3) which only provides examples like (+1+1), (+1,-1), (-1,-1)? The theoretical proof does not go through without unbiased risk estimators, which in this is obtainable because of Thm 1 and 2, but these are very specific to the above assumption on the generative model? But I think it is an okay assumption and the bare minimum for any practical purpose.

Also, can you comment on the tightness of the current bound say Thm. 4 in terms of n (for this specific generative model).

Overall I found the paper is easy to understand and well structured.

Two interesting follow-up questions:
- It would interesting if the authors can add some comments about the possibility of extending the work to a multiclass classification setting.
- Another interesting direction could be to understand the performance limits for the case where instead of two the learner is allowed to see the relative class probabilities of a larger subset of items. Is the current method directly extendable to such a setting and how do we expect to see the estimation error bound varying with the subsetsize?

---

> ### Author Response · Authors · 2020-11-18
> **Response to Reviewer #4**
>
> Thank you for your insightful comments!
>
> Q1. The data generation model only provides examples like $(+1, -1), (+1, -1), (-1, -1)$?
>
> A1. Yes, while a pair of $(x, x')$ with labels $(-1, +1)$ would not be discarded. Specifically, for a pair of $(x, x')$ with labels $(-1, +1)$, $(x', x)$ could be taken as a pairwise comparison example. Because by exchanging the positions of $(x, x')$, $(x', x)$ would be associated with labels $(+1, -1)$, which belong to the above three cases. We adopt such an assumption for the data generation process because it is much harder to obtain the positive confidence $p(y=+1|x)$ than the label $y$ for each example $(x, y)$.
>
> Q2. Unbiased risk estimators are specific to the above assumption on the generative model?
>
> A2. Yes, the derived unbiased risk estimators are based on the above assumption on the data generation model. If we adopt another reasonable assumption on the data generation model, we may derive another unbiased risk estimator, which would also hold an estimation error bound.
>
> Q3. Can you comment on the tightness of the current bound say Thm. 4 in terms of $n$?
>
> A3. Thank you for the nice suggestion. As we can see, the estimation error bound gets tighter with $n$ increasing. It is also worth noting that the Rademacher complexity of the model class can normally be bounded by $C/\sqrt{n}$ for a positive constant $C$. Hence, we can further see that the convergence rate is $\mathcal{O}_p(1/\sqrt{n})$ where $\mathcal{O}_p$ denotes the order in probability. This order is the optimal parametric rate for empirical risk minimization without additional assumptions (Mendelson, 2008).
>
> Shahar Mendelson. Lower bounds for the empirical minimization algorithm. IEEE TIT 2008.
>
> Q4. The possibility of extending the work to a multiclass classification setting.
>
> A4. Thank you for your wonderful advice. We could extend Pcomp classification to the multi-class classification setting by using the one-versus-all strategy. Suppose there are multiple classes, we are given pairs of unlabeled data that we know which one is more likely to belong to a specific class. Then, we can use the proposed methods in this paper to train a binary classifier for each class. Finally, by comparing the outputs of these binary classifiers, the predicted class can be determined. We have added the above discussions to Conclusion.
>
> Q5. Is the current method directly extendable to such a setting (a larger subset of examples with relative class probabilities instead of two) and how do we expect to see the estimation error bound varying with the subset size?
>
> A5. Thank you for your nice questions. Our methods may not be directly extendable to such a setting, while by decomposing a larger subset of examples with relative class probabilities into pairwise comparison examples, we can still use our methods to solve this problem. In this case, we can also expect that the estimation error bound gets tighter as the subset size gets larger, because a larger subset will result in more pairwise comparison examples.

---

### Official Review · AnonReviewer3 · 2020-10-31
**problem is not novel**

**Rating:** 4
**Confidence:** 3

**Review:**

The paper develops a method to learn a binary classifier based only on pairwise comparison data. For example, the classifier learns to classify pictures of people as "adult" versus "child" based on pairwise comparisons of the form "person C is older than person X". The authors derive their method based on an empirical risk minimization argument. The authors test their methods on four standard data sets (three MNIST variants and one more). They compare to some baselines including binary biased, noisy unbiased, and RankPruning. They try 4 different variations of their method. The Pcomp-Teacher model performs especially well.

The paper develops some interesting ideas, but the main problem is that the authors say that their pairwise comparison (Pcomp) classification is novel, but it is not. For example, see this blog post: https://blog.ml.cmu.edu/2019/03/29/building-machine-learning-models-via-comparisons/

And this paper:

Noise-tolerant interactive learning using pairwise comparisons.
Yichong Xu, Hongyang Zhang, Kyle Miller, Aarti Singh, Artur W Dubrawski. NIPS'17: Proceedings of the 31st International Conference on Neural Information Processing Systems December 2017 Pages 2428–2437
https://dl.acm.org/doi/10.5555/3294996.3295004

As the blog post points out, learning to rank is very related and there are hundreds of papers on learning to rank.

Usually pairwise comparisons are motivated as easier for people to do (lower cost labels to obtain). For example, it's easier to tell if one person is older than another than to guess the age of a person. The authors motivate pairwise comparison differently. They say it is better for privacy. Why are pairwise comparisons better for privacy? Given enough pairwise comparisons, the original order can be uncovered, so there are limits to the privacy advantage that the authors don't discuss. In general, the privacy motivation needs to be more clear.

Pairwise can sometimes be harder to judge than pointwise, for example which of two pictures of a cat is "more" cat than the other? Or which of two laptop computers is "more cat-like" than the other?

The authors must get some small amount of pointwise data, or they have some assumption that I missed, like the base label frequencies are known. For example, what if we try to learn a dog/cat classifier using pairwise comparisons that are 100% cats? How can the classifier possibly know that all the images are cats without *some* pointwise labels? All we know from pairwise comparisons is which cat is the most and least dog-like. There is no way to know how to draw the line without some pointwise labels or other modeling assumption. The blog post above discusses this.

---

> ### Author Response · Authors · 2020-11-18
> **Response to Reviewer #3**
>
> Thank you for your constructive comments!
>
> Q1. Novelty of Pcomp classification.
>
> A1. Thank you for pointing out the related pairwise comparison work (Xu et al., 2017). We have cited it and discussed the difference between it and our work in Section 2.
>
> We argue that our proposed pairwise comparison (Pcomp) classification is indeed novel:
>
> a) Problem setting: In Xu et al. (2017), they interactively learn a binary classifier from (noisy) labeling and comparison oracles. They need both pairwise comparison data and pointwise labeled data, while Pcomp classification only requires pairwise comparison data, without any pointwise labeled data. This is a significant difference that makes the two problems fairly different (though related). To the best of our knowledge, we are the first to study the Pcomp classification problem where no pointwise labeled data are provided.
>
> b) Method: Learning from only pairwise comparison data is much more difficult, which is not an incremental step over the problem studied by previous works (Xu et al., 2017; Kane et al., 2017). Specifically, previous methods cannot be used to solve our problem, but our proposed methods can be easily adapted to solve the problem where extra labeled examples are provided (by further incorporating the training loss on labeled examples into the objective function). Unlike previous studies (Xu et al., 2017; Kane et al., 2017) that leverage some pointwise labels to differentiate the labels of pairwise comparison data, we are the first to learn from only pairwise comparison data in the framework of empirical risk minimization. Therefore, our methods are compatible with any deep network architecture / stochastic optimization, and hold an estimation error bound with optimal convergence rate.
>
> Yichong Xu, Hongyang Zhang, Kyle Miller, Aarti Singh, Artur Dubrawski. Noise-tolerant interactive learning from pairwise comparisons with near-minimal label complexity. NeurIPS 2017.
>
> Daniel M. Kane, Shachar Lovett, Shay Moran, Jiapeng Zhang. Active classification with comparison queries. FOCS 2017.
>
> Q2. Relationship with learning to rank.
>
> A2. Thank you for raising this concern. Learning to rank is conceptually related to our paper (due to pairwise comparison data), while they are essentially different:
>
> Our goal is binary classification, rather than ranking. Even if we have learned a perfect ranking function (we know for any pair of examples that one is more positive than the other), we still cannot use it to conduct effective binary classification.
>
> Unlike the method proposed by Xu et al. (2017) that leverages the ranking among training examples to differentiate negative and positive examples by a changing point, our proposed methods directly work by empirical risk minimization, which do not rely on a ranking among training examples.
>
> Q3. Pairwise comparisons are helpful for cost-saving labeling, not for data privacy.
>
> A3. Thank you for pointing out this issue. We agree with you that pairwise comparisons are easier for people to collect. We would like to express that especially for private matters, pairwise comparisons are easier to collect than pointwise labels. For example, it is impolite to ask a lady’s age (private matter), while it is easier to know which person is younger than the other. We have revised Abstract and the second paragraph in Introduction, to clearly state that pairwise comparisons are easier for people to collect than pointwise labels.
>
> Q4. Pairwise can sometimes be harder to judge than pointwise.
>
> Q4. Yes, we agree with you that pairwise can sometimes be harder to judge than pointwise. Besides, it is extremely difficult to obtain the exact positive confidence of each example. Therefore, we assumed a generation process of pairwise comparison examples based on the labels (Section 3), instead of comparing the positive confidences of two examples.
>
> Q5. The authors must get some small amount of pointwise data, or they have some assumption that I missed.
>
> A5. Thank you for raising this concern. We do not need any pointwise labeled data, while our methods rely on the assumption about the generation process of pairwise comparison data (Section 3). With the data generation process, we will have pairwise comparison data that come from both positive and negative classes, thereby training an effective classifier.

---

### Author Response · Authors · 2020-11-19
**Summary of revisions**

We sincerely appreciate all reviewers for their valuable and constructive comments, which make our paper better without doubt.

We have meticulously addressed their mentioned problems and revised our manuscript based on their valuable suggestions. In particular, we have made the following key changes:

- We have revised one sentence in Abstract and the second paragraph in Introduction to clearly state that pairwise comparisons are usually easier for people to collect than pointwise labels (suggested by Reviewer #3).

- We have added several sentences in the first paragraph in Section 2 to discuss related studies (pointed out by Reviewer #1 and Reviewer #3).

- We have provided more detailed comments on the estimation error bounds (suggested by Reviewer #4).

- We have discussed in Conclusion how we could extend Pcomp classification to the multi-class classification setting (suggested by Reviewer #4).

- We have revised the second and the third paragraph in Section 3 to make the data generation process clearer (suggested by Reviewer #1).

---

### Decision · Program_Chairs · 2021-01-07
**Final Decision**

**Decision:**

Reject

**Comment:**

This paper has been evaluated  by three expert reviewers, two of whom recommended rejection and one acceptance. Two of the three reviews are particularly detailed and thorough. Both point out a few points of conceptual issues that leave the reader confused. These key issues have not been addressed sufficiently in the rebuttal to result in changing the reviewers' assessments. One major concern is lacking novelty of the work as presented, which limits its current utility to the ICLR audience. I recommend a rejection.